# Influence of Internal Structure of the Sorbents on Diazepam Sorption from Simulated Intestinal Fluid

**Mircea Stefan** [1,2], **Ioana Stefan** [3], **Ioana-Alexandra Negoita** [4], **Viorel Ordeanu** [2] and **Daniela Simina Stefan** [1,*]

1   Faculty of Applied Chemistry and Materials Science, University Politehnica from Bucharest,
    No. 1-7, Gh. Polizu Str., Sector 1, 011061 Bucharest, Romania; mircea.stefan@upb.ro
2   Pharmacy Faculty, University Titu Maiorescu, No. 22 Dâmbovnicului Str., Secor 4,
    040441 Bucharest, Romania; viorel.ordean@prof.utm.ro
3   Pharmacy Faculty, University of Medicine and Pharmacy Carol Davila, Bucharest, 37 Dionisie Lupu Str.,
    020021 Bucharest, Romania; ioana.stefan@rez.umfgc.ro
4   Medicine Faculty, University of Medicine and Pharmacy Carol Davila, 8 Eroii Sanitar Bd.,
    020021 Bucharest, Romania; alexandra.naftanaila@doc.umfcd.ro
*   Correspondence: simina_stefan_ro@yahoo.com

**Featured Application: Activated charcoal and bentonite could be used for the treatment of the poisoned persons especially as a household remedy.**

**Abstract:** The capacity of natural Na-montmorillonite and activated charcoal for sorption of diazepam from simulated intestinal fluid (SIF) was studied. The main characteristics of the sorbents were determined. In order to characterize the sorption process of diazepam the influence of the pH, contact time and ethanol presence in SIF was analyzed. Adsorption isotherms for the diazepam-activated charcoal and diazepam-natural Na-montmorillonite systems were determined. The Langmuir isotherm model provided a very good description of diazepam sorption. Furthermore, the pH-drift method was used to determine the specific pH at zero point of charge (pHzpc) of the sorbents. The obtained results show that the internal structure of the sorbents and pH of the SIF solutions are very important for diazepam sorption. Both the surface of the activated charcoal and natural Na-montmorillonite are positively charged below the pHzpc so the sorption of diazepam is higher below this point and occur by van der Waals forces. The presence of ethanol in simulated intestinal fluid lowers the adsorption of diazepam on both sorbents.

**Keywords:** diazepam; accidental poisoning; sorption; simulated intestinal fluid; activated charcoal; montmorillonite





## 1. Introduction

Suicidal or accidental poisoning is considered as a major problem in many countries. In the last years, in Romania was developed a research regarding the drug related deaths. The toxicological analysis indicated the presence of barbiturates, benzodiazepines, cannabinoids and opiates in about 55% of cases [1].

For example, in the United States in 2018, about 67,500 drug overdose deaths occurred. The age-adjusted rate of overdose deaths decreased by 4.6% from 2017 (21.7 per 100,000) to 2018 (20.7 per 100,000) [2]. The chemical compounds used for self-poisoning include drugs (benzodiazepines, paracetamol, phenytoin, fluoxetine and amitriptyline) nicotine, pesticides (organophosphates), rodenticides (zinc phosphide), opioids, cannabinoids, stimulants, volatile solvents, hallucinogens and alcohol [3]. The type of drug, the amount taken and the health form of the person who overdosed determine the severity of a drug poisoning.

A recent study indicates that the number of intentional poisoning cases is higher than accidental poisoning. Additionally, sedative and hypnotics were found to be the most

common types of drugs (about 43%) used for intentional suicidal attempt [4]. Many studies have shown benzodiazepines as one of the major drugs responsible for self-poisoning throughout the world [3–5]. Among the benzodiazepines, diazepam is one of the most widely used drugs for different clinical purposes such as anxiety, acute alcohol withdrawal, status epilepticus and other convulsive states. Because of wider therapeutic uses, diazepam is also the most common benzodiazepine in poisoning [5].

Oral benzodiazepine (BZD) overdoses, without co-ingestions, rarely result in significant morbidity (e.g., aspiration pneumonia, rhabdomyolysis) or mortality. In mixed overdoses, they can potentiate the effect of alcohol or other sedative-hypnotics. The clinical manifestations of benzodiazepine overdose include slurred speech, ataxia, altered mental status and respiratory and central nervous system (CNS) depression [5]. The intentional ingestions of benzodiazepines frequently involve coingestants such as ethanol or other drugs, ethanol being the most common. Because ingestion depends on multiple factors including weight, age tolerance and coingestants, it is difficult to quantify the amount of diazepam dose necessary to produce respiratory depression. It must be noted that patients with severe toxicity will present stuporous or comatose [6].

Activated charcoal and bentonite are frequently used for the treatment of the poisoned persons especially as a household remedy [7]. Activated charcoal (AC) is obtained by wood or other carbonaceous material pyrolysis, followed by the oxidation of the pyrolyzed product with carbon dioxide or steam. The surface area of the resulted material is in range of few hundred up to two thousand $m^2$ per gram [8,9]. Usually, the adsorption capacity of activated charcoal is determined by its internal pore volume. The adsorption process depends on the pore size and the type of activation of carbon. Moreover, the surface of activated charcoal contains several moieties that adsorb drugs and poisons to varying degrees [10]. Activated charcoal can adsorb onto its surface chemicals, medical drugs and phytotoxins, preventing their absorption from the gastrointestinal tract [11].

Smectites are clay minerals that possess physiochemical properties that have made it possible to be used in pharmaceutical industry. Adsorption capacity, swelling capacity, chemical inertness, high specific area, favorable rheological properties and colloidal properties are few of the most important characteristics of clay minerals which give them importance for pharmaceuticals [9–12]. The most common smectite is montmorillonite, a mineral clay with a structure made by layers resulted from condensation of two tetrahedral sheets and one central octahedral sheet. In the presence of water, the volume of montmorillonite can expand several times [13].

The substitution of few octahedral $Al^{3+}$ with $Mg^{2+}$ occur in montmorillonite and it is responsible for its negative charge which is equilibrated by the cations sorbed on basal planes. These cations form outer-sphere surface complexes [14] and can be exchanged with other cations existing in solute. Furthermore, montmorillonite is characterized by pH-dependent sorption properties [7].

In this study we have investigated two sorbents, Na-montmorillonite and activated charcoal, as an antidote for diazepam poisoning.

## 2. Materials and Methods

### 2.1. Materials Characterisation

The activated charcoal used in this study was a activated coconut charcoal powder supplied by SORBOTECH-Poland. Activated charcoal has a cation exchange capacity (CEC) of 2.75 meq/g and 93.1% of the particles <10 μm.

The natural Na-montmorillonite used for sorption experiments was provided by MINESA-S.A. Cluj Napoca. Natural Na-montmorillonite has a CEC of 0.93 meq/g and 92% of the particles <2 μm.

Before the beginning of the experiments both activated charcoal and Na-montmorillonite was oven-dried at 105 °C until all the moisture has evaporated (constant mass). Each sorbent was stored in a desiccator at room temperature (22.0 ± 0.5 °C).

In Table 1 are presented information regarding diazepam.

**Table 1.** Characteristics of drug molecule.

| Formula | Chemical Structure Image, (2D) | Molar Mass (g·mol$^{-1}$) | Product | Supplier |
|---|---|---|---|---|
| $C_{16}H_{13}ClN_2O$ |  | 284.74 | Diazepam (10 mg) | Terapia-RANBAXY |

The stock diazepam solution was prepared by dissolving the drug in 500 mL of simulated intestinal fluid. The ethanol used in this study was 96% ethanol supplied by Sigma-Aldrich 9630.

We have used pH-drift method in order to calculate pH at zero point of charge (pHzpc) [7].

Simulated Intestinal Fluid Preparation

Simulated intestinal fluid (SIF—pH 6.8) without pancreatin was prepared according to the United States Pharmacopeia, Ref. [15]. 6.8 g of $KH_2PO_4$ were dissolved into 50 mL of water. Then were added 190.0 mL of 0.2N NaOH and deionized water to make 1000 mL SIF.

*2.2. Sorbents Characterization*

In Table 2 are presented the methods and apparatus used to characterization of sorbents

**Table 2.** Methods to characterization of sorbents.

| No. Crt. | Method Used | Characteristics Obtained | Apparatus Used |
|---|---|---|---|
| 1 | Brunauer–Emmett–Teller (BET) equation | Pore size and surface area | Micromeritics TriStar II 3020 Analyzer |
| 2 | X-ray fluorescence (XRF) analysis | Chemical composition of natural Na-montmorillonite | JASCO FP-6500 Analyzer |
| 3 | X-ray diffraction (XRD) analysis | X-ray diffraction (XRD) pattern of natural Na-montmorillonite. | Shimadzu XRD 6000 diffractometer with Ni filtered Cu Kα radiation (λ = 1.5406 Å) |
| 4 | Fourier transform infrared spectroscopy (FT-IR) | IR Spectrum of mineral clay and activated charcoal samples | Perkin-Elmer Spectrum 100 spectrometer with ATR, wave number range 4000–400 cm$^{-1}$ |
| 5 | Scanning electron microscopy (SEM) | SEM images for activated charcoal and mineral clay | HITACHI S2600N |

Pore Size and Surface Area

Pore size and surface area were determined both for activated charcoal, and mineral clay by $N_2$ adsorption techniques. Brunauer–Emmett–Teller (BET) equation was used to calculate the BET surface area from the isotherms [16]. Using the value of amount of adsorbed nitrogen at P/Po 0.95, the total pore volume was calculated. Dubinin–Radushkevich equation was used to determine the micropore volume [17]. The value of the amount of $N_2$ adsorbed between relative pressures P/Po 0.40–0.95 was used to calculate mesopore volume. The molar volume of liquid nitrogen was considered of 35.0 cm$^3$/mol [18]. The pore size distribution is ascertained by non-local density functional theory (NLDFT) [19].

In a previous study we have determined pore size distribution, pore volume and surface area of natural Na-montmorillonite [7].

### 2.3. Influence of Contact Time on Diazepam Sorption

### 2.3.1. Influence of Initial Drug Concentrations

Fifty milligrams of sorbent were contacted with 50 mL of diazepam SIF solutions in glass vessels immersed in a thermostatic water bath, at $37.0 \pm 0.1$ °C. Small volumes of $10^{-2}$ M $HNO_3$ or $10^{-2}$ M NaOH solutions were added, in order to maintain the suspensions pH at $6.8 \pm 0.05$. The pH was measured by Agilent 3200 pH meter. We have calibrated the pH electrode with buffers (Merk, titrisol) at 37 °C. Rotating magnetic bar ensured a good stirring of the suspension. Then, 5 mL samples were withdrawn from the reaction vessel at given times and immediately centrifuged.

For activated charcoal, the initial concentration of diazepam SIF solutions were 0.3, 0.5 and 0.7 g/L and for natural Na–montmorillonite was 6, 12 and 18 mg/L.

A high-performance liquid chromatography (HPLC) system was used to determine the diazepam concentrations, both before and after adding activated charcoal or Na-montmorillonite. The HPLC system included: Integrator Model CR501, liquid pump Model LC-10AS, variable wavelength UV-VIS detector Model SPD-10A, system controller Model SCL-10A and auto-injector Model SIL-10A.

A Hypersil reversed phase C-18 column with 250 by 4.6 mm i.d. and 5 μm particle size was used. The mobile phase has contained acetonitrile methanol, 1% phosphate buffer (pH—3) in ratio of 18:58:24 (v/v/v). The flow rate was 1 mL/min. The eluent was monitored by UV detection at 232 nm. By plotting average peak area against concentration calibration curve was created.

We have calculated the adsorbed diazepam amount by means of the difference between the initial and the final concentration of the solution.

### 2.3.2. Influence of Sorbent Particle Sizes

Activated charcoal sorption experiments occurred in the same way as described before, but for various sizes of particle (D < 10 μm, 10–15 μm and 15–18 μm, respectively) and initial diazepam SIF solution concentration of 0.7mg/mL.

Additionally, three different particle sizes of natural Na-montmorillonite (D < 2 μm; 4 μm < D < 6 μm; 8 μm < D < 10 μm) were used for sorption of diazepam. In this case, the initial diazepam SIF solution concentration was 18 mg/L [7].

### 2.4. Sorption Equilibrium of Diazepam

Twenty milligrams of sorbent were contacted for 2 h with 20 mL of diazepam SIF solutions in glass vessels immersed in a thermostatic water bath, at $37.0 \pm 0.1$ °C until equilibrium attainment. Small volumes of $10^{-2}$ M $HNO_3$ or $10^{-2}$ M NaOH solutions were added in order to maintain the suspensions pH at $6.8 \pm 0.05$. For sorption on activated charcoal, the initial diazepam SIF solutions concentrations ranged from 0.2 to 0.8 mg/mL while for sorption on natural Na-montmorillonite the initial diazepam concentrations ranged from 2 to 20 mg/L.

After equilibrium attainment, suspensions were centrifuged (Universal 32 Hettich) for 30 min at 10,000 rpm. HPLC system was used to determine the diazepam concentrations in the supernatant.

### 2.5. Effect of pH on Diazepam Sorption

Then, 20 mL of diazepam SIF solutions were contacted for 2 h with 20 milligrams of sorbent in glass vessels immersed in a thermostatic water bath; at $37.0 \pm 0.1$ °C. Small volumes of $10^{-2}$ M $HNO_3$ or $10^{-2}$ M NaOH solutions were added in order to vary the pH in range 6 to 9.5. For sorption on activated charcoal the initial diazepam SIF solution concentration was 0.7 mg/mL, while for sorption on natural Na-montmorillonite, the initial drug concentration was 18 mg/L. Agilent 3200 pH meter was used for pH measuring.

After equilibrium attainment, suspensions were centrifuged for 30 min at 10,000 rpm, and HPLC system was used to determine the diazepam concentrations in the supernatant.

### 2.6. The Influence of Ethanol on Diazepam Sorption

The influence of ethanol on diazepam sorption from SIF-ethanol solutions was described using a Langmuir isotherm. Three SIF-ethanol solutions with ethanol concentrations of 5%, 10% and 15% (volume ethanol: Volume SIF) were analyzed.

The sorption experiments were carried out in the same way as described in Section 2.4.

The glass electrode of the pH meter has been calibrated in each SIF-ethanol mixtures (5%, 10% and 15% ethanol). Additionally, for each SIF-ethanol mixtures HPLC calibration curve was constructed by plotting average peak area against concentration and regression equation was computed.

For all experiments the solid: Liquid ratio was of 1 g/L. All experiments were realized in duplicate.

## 3. Results

### 3.1. Sorbents Characterization

3.1.1. Pore Size Distribution and Specific Surface Area

Particle size distribution, pore volume and pore size distribution that characterize each sorbent are presented in Tables 3–6.

**Table 3.** Activated charcoal particle size distribution.

| Fraction | % |
|:---:|:---:|
| <10 μm | 93.1 |
| 10–15 μm | 2.2 |
| 15–18 μm | 4.7 |

**Table 4.** Activated charcoal pore size distribution.

| Pore diameter | Pore distribution, % | | |
|:---:|:---:|:---:|:---:|
| 5–10 Å | 0 | | |
| 10–15 Å | 28.35 | | |
| 15–25 Å | 39.86 | Pore volume (0–100 Å), $cm^3 \cdot g^{-1}$ | 1.132 |
| 25–50 Å | 20.41 | | |
| 50–100 Å | 11.38 | | |
| 100–300 Å | 0 | | |
| 300–1000 Å | 0 | | |
| 1000–10,000 Å | 0 | Pore volume (100–75,000 Å), $cm^3 \cdot g^{-1}$ | 0 |
| 10,000–75,000 Å | 0 | | |

**Table 5.** Natural Na-montmorillonite particle size distribution [7].

| Fraction | % |
|:---:|:---:|
| <2 μm | 92 |
| 4–6 μm | 3.9 |
| 8–10 μm | 4.1 |

**Table 6.** Pore size distribution of natura Na-montmorillonite [7].

| Pore diameter | Pore distribution, % | | |
|---|---|---|---|
| 5–10 Å | 0 | | |
| 10–15 Å | 9.67 | | |
| 15–25 Å | 25.85 | Pore volume (0–300 Å), $cm^3$ $g^{-1}$ | 0.153 |
| 25–50 Å | 28.41 | | |
| 50–100 Å | 10.81 | | |
| 100–300 Å | 25.26 | | |
| 300–1000 Å | 0 | | |
| 1000–10,000 Å | 0 | Pore volume (300–75,000 Å), $cm^3$ $g^{-1}$ | 0 |
| 10,000–75,000 Å | 0 | | |

### 3.1.2. XRF Characterization

The chemical composition of the natural Na-montmorillonite is presented in Table 7.

**Table 7.** Chemical analysis of natural mineral clay [7].

| Chemical Composition | $ZrO_2$ | $TiO_2$ | CaO | $K_2O$ | $Fe_2O_3$ | $Na_2O$ | $Al_2O_3$ | $SiO_2$ |
|---|---|---|---|---|---|---|---|---|
| Raw clay weight % | 0.083 | 0.607 | 1.94 | 1.97 | 3.37 | 7.33 | 13.4 | 71.3 |

### 3.1.3. XRD Characterization

The natural Na-montmorillonite mineralogical composition was evaluated by using of X-ray diffraction (XRD). The composition of mineral clay was determined by comparing "d" values and consist in calcite (3.05 Å), kaolinite (7.24 Å) and montmorillonite (12.3 Å and quartz (3.31 Å) [7].

### 3.1.4. FTIR Characterization

The vibrational spectrum of the sorbents was investigated using FTIR spectroscopy technique, in order to determine characteristics of each surface. The vibrational spectrum of activated charcoal is presented in Figure 1.

The broad absorption band at 3600–3200 $cm^{-1}$, with a maximum at about 3440 $cm^{-1}$, due to the O\H stretching $\nu$(O\H) indicate the presence of hydroxyl groups from alcohols, phenols or carboxyls, adsorbed in the activated carbons. The band at 2800–3000 $cm^{-1}$ could be assigned to stretching of methyl groups $\nu$(C\H). The spectrum shows a pronounced band at 1631 $cm^{-1}$, that reveals the C=C stretching vibration in the structure of the activated carbon. The band at 1000–1300 $cm^{-1}$ usually has been assigned to C-O stretching in alcohols, phenols, acids, ethers and esters groups [20,21] The peak at 1162 $cm^{-1}$ indicates the stretching mode of hydrogen-bonded P = OOH groups from phosphates or polyphosphates. The peak at 1081 $cm^{-1}$ could be determined by the symmetrical vibration in polyphosphate chain P–O–P and $P^+$–$O^-$ in acid phosphate esters [22]. The FT-IR spectroscopy result indicates that the activated charcoal is rich in surface functional groups.

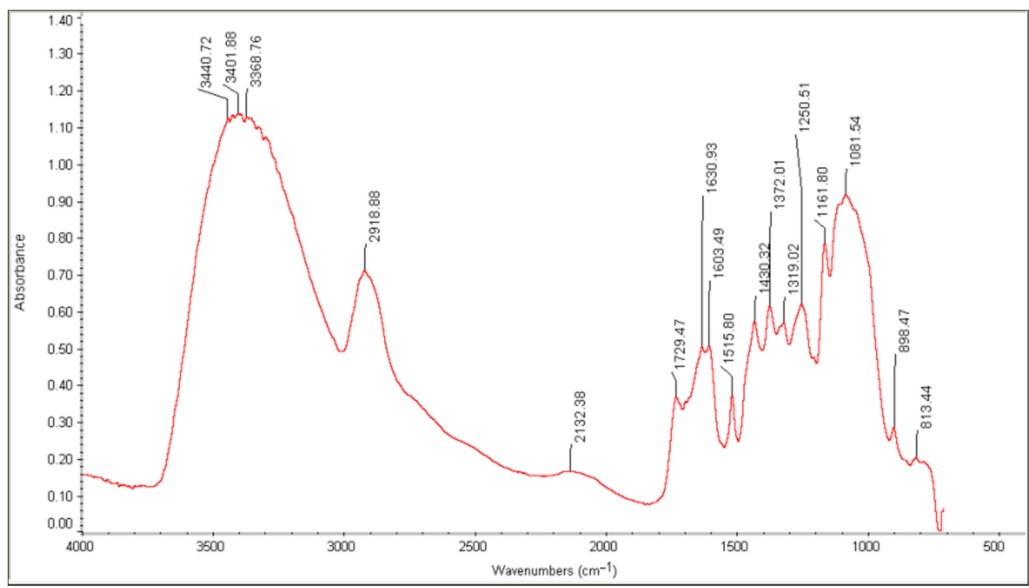

**Figure 1.** FT-IR spectrum of the activated charcoal.

The vibrational spectrum of natural Na-montmorillonite is illustrated in Figure 2.

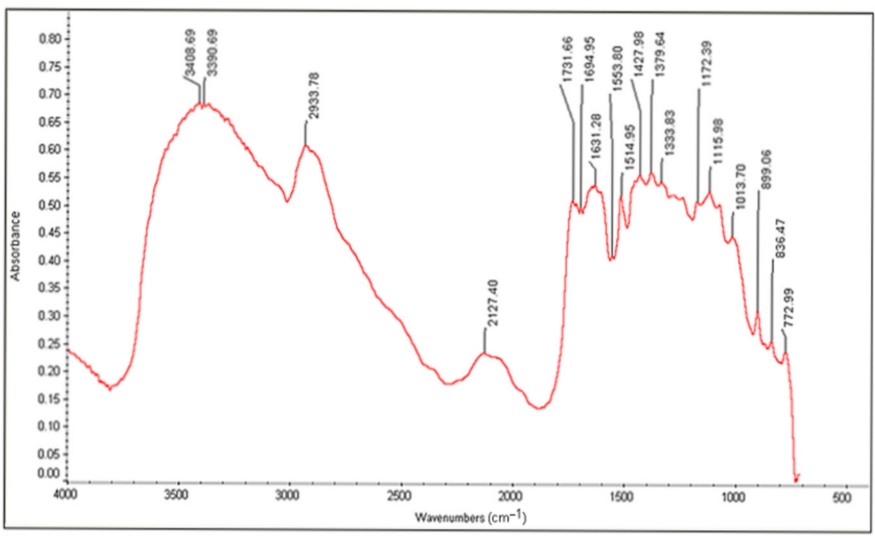

**Figure 2.** FT-IR spectrum of the natural Na-montmorillonite.

The presence of hydroxyl groups from Fe-OH-Al and Al-OH-Al is indicated by the OH broadband which exhibits the maximum at the wave number of approximatively 3409 cm$^{-1}$. The presence of bending H-O-H vibration is suggested by the band that arises at 1632 cm$^{-1}$. At the wave number of approximatively 899 cm$^{-1}$, is observed the band corresponding to Al-Al-OH. Absorption bands at 773, 1116 and 1013 cm$^{-1}$ originating from the stretching and bending vibrations of $SiO_2$ tetrahedral indicate that the bending mode of Si-O is present in the silicate structure [23]. SiO stretching vibrations observed at 773 cm$^{-1}$ support the presence of quartz. The wave number of approximatively 1428 cm$^{-1}$ indicate the presence of calcium combined with carbonate species and reveals the existence of calcite [7].

### 3.1.5. SEM Analysis

Figure 3 illustrates SEM micrographs of natural Na-montmorillonite and suggests the presence of clay particles with a diameter in the range of 1.5 μm. The predominance of

montmorillonite, that was shown by XRD technique is confirmed by SEM examination. The technique confirmed fluffy or open-textured of Na-montmorillonite which is supposedly because the Na interlayer cation is more easily hydrated.

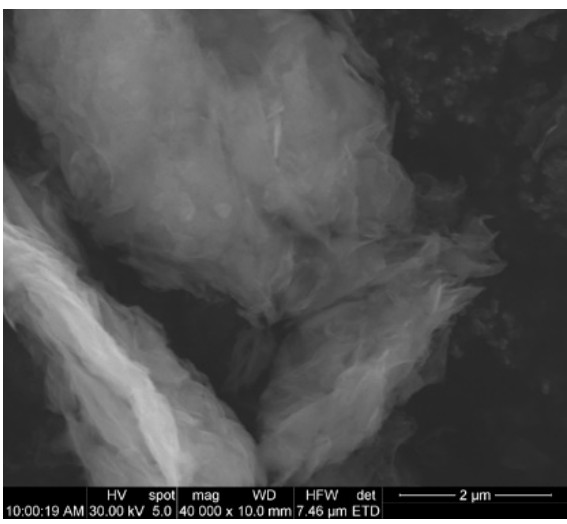

**Figure 3.** SEM micrographs of mineral clay.

Figure 4 shows a smooth activated charcoal surface and a visible presence of pores. The pores on the activated charcoal surface were uniform in the micropore to mesopore size range. The SEM analysis confirms the presence of micropores and mesopores that were identified by $N_2$ adsorption techniques.

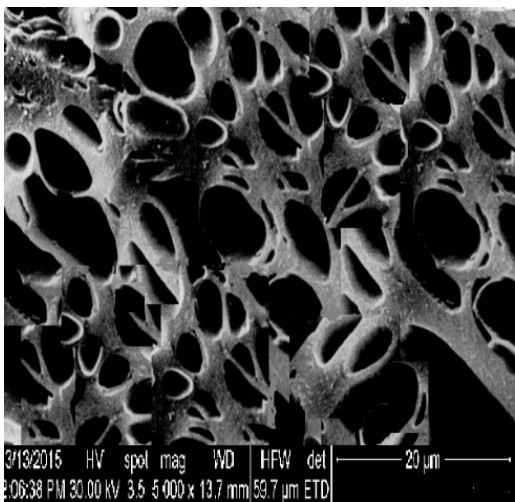

**Figure 4.** SEM micrographs of activated charcoal.

### 3.2. Influence of Contact Time on Diazepam Sorption

The influence of the contact time on diazepam uptake by activated charcoal and natural mineral clay at various initial concentrations and for various sorbent particle sizes is presented in Figures 5–8 and indicate that drug sorption equilibrium is reached within 2 h. The mass of drug uptake after 4 h were not significantly different from the mass adsorbed after 2 h.

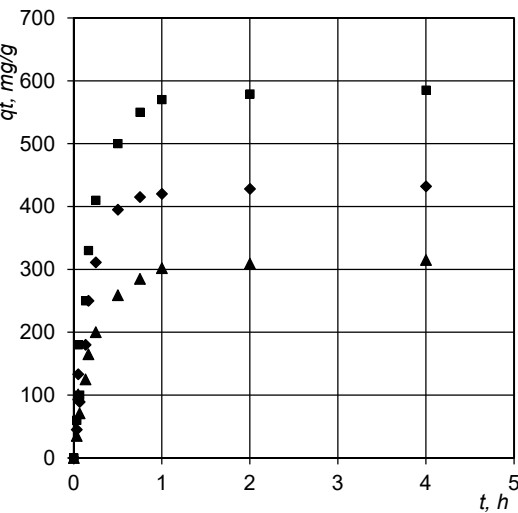

**Figure 5.** The influence of contact time on diazepam concentration uptake by activated charcoal, for different initial concentrations; $T$ = 37.0 ± 0.1 °C; pH = 6.80 ± 0.05; ■ $C_i$ = 0.7 mg/mL; ◆ $C_i$ = 0.5 mg/mL; ▲ $C_i$ = 0.3 mg/mL; and S:L ratio = 1 g/L.

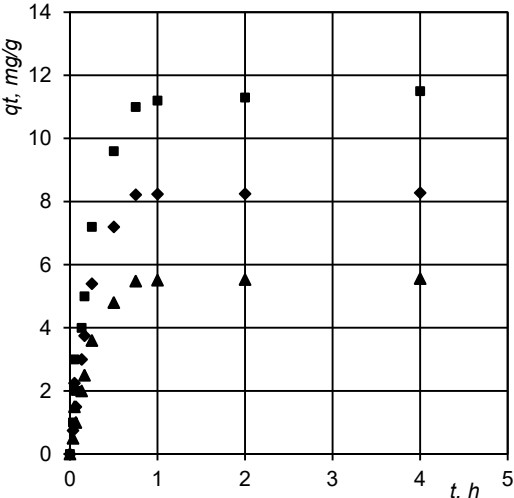

**Figure 6.** The influence of contact time on diazepam concentration uptake by mineral clay, for different initial concentrations; T = 37.0 ± 0.1 °C; pH = 6.80 ± 0.05; ■ $C_i$ = 18 mg/L; ◆ $C_i$ = 12 mg/L; ▲ $C_i$ = 6 mg/L; and S:L ratio = 1 g/L.

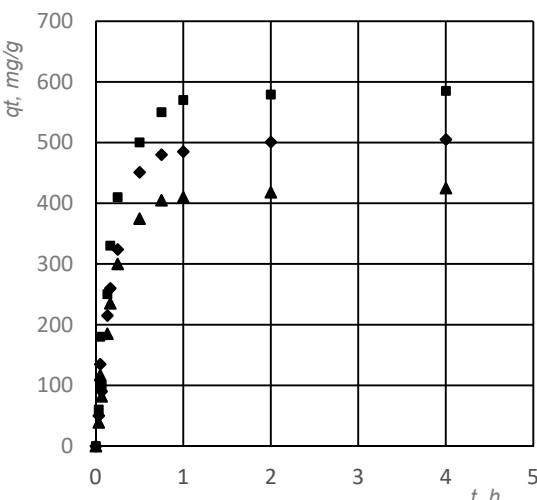

**Figure 7.** The influence of contact time on diazepam concentration uptake by activated charcoal for different sorbent particle sizes; $T = 37.0 \pm 0.1$ °C; pH = $6.80 \pm 0.05$; $C_i = 0.7$ mg/mL; ■ D < 10 μm; ◆ 10 μm < D < 15 μm; ▲ 15 μm < D < 18 μm; and S:L ratio = 1 g/L.

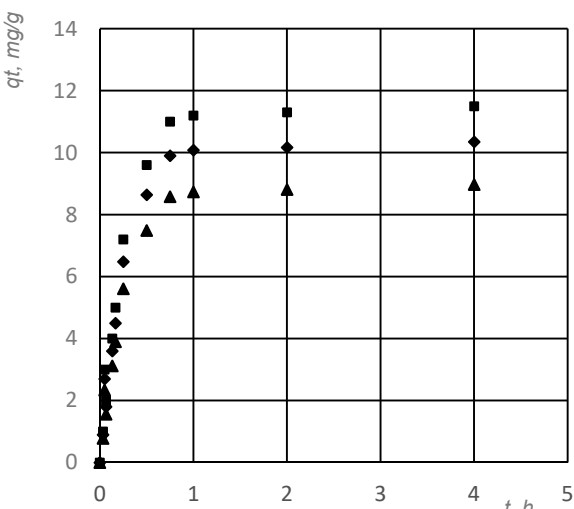

**Figure 8.** The influence of contact time on diazepam concentration uptake by mineral clay for different sorbent particle sizes; $T = 37.0 \pm 0.1$ °C; pH = $6.80 \pm 0.05$; $C_i = 18$ mg/L; ■ D < 2 μm; ◆ 4 μm < D < 6 μm; ▲ 8 μm < D < 10 μm; and S:L ratio = 1 g/L.

The mass, $q_t$, of drug uptake at time $t$, was determined from the equation:

$$q_t = (C_0 - C_t) \ V/m \tag{1}$$

where: $C_0$—initial drug concentration (mg/L); $C_i$—final drug concentration (mg/L); $V$—volume of solution (L); and $m$—mass of sorbent (g).

### 3.2.1. Influence of Initial Drug Concentrations

The dependence of diazepam concentration uptake by sorbents on the contact time is presented in Figures 5–8. An increasing of the initial drug concentration determines an increasing of the amount of drug adsorbed.

### 3.2.2. Influence of Sorbent Particle Sizes

The influence of the sorbent particles size was analyzed. As illustrated in Figures 7 and 8, the decreasing of both sorbent's particles size determines increasing of the amount of drug uptake.

The mass, $q_t$, of drug uptake at time $t$, is calculated from Equation (1).

### 3.3. Sorption Equilibrium of diazepam

Equilibrium isotherm equations are used to describe adsorption experiments. One of most common isotherms used to describe solid-liquid sorption is the Langmuir isotherm.

Langmuir isotherm can be expressed as

$$a = a_m \frac{bC}{1 + bC} \tag{2}$$

where: $a$—equilibrium concentration of adsorbed drug (mg·g$^{-1}$), $C$—equilibrium concentration of drug in solution (mg·L$^{-1}$); $a_m$—the maximum concentration of adsorbed drug (mg·g$^{-1}$); and $b$—equilibrium constant (L·mg$^{-1}$)

By linearization, Equation (2) can be written as:

$$\frac{C}{a} = \frac{C}{a_m} + \frac{1}{a_m b} \tag{3}$$

Thus, a plot of C/a vs. C will provide a straight line of slope $1/a_m$ and intercept $1/ba_m$. Diazepam adsorption isotherms are shown in Figures 9 and 10.

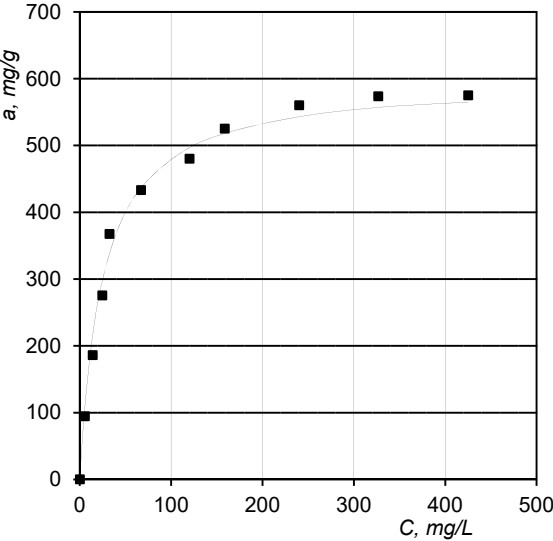

**Figure 9.** Diazepam adsorption isotherm for activated charcoal; $T$ = 37.0 ± 0.1 °C; pH = 6.80 ± 0.05; S:L ratio = 1 g/L; ■ experimental data; and—Langmuir-fit adsorption curves.

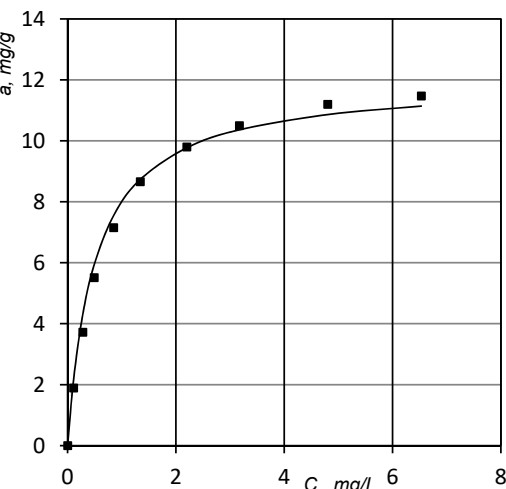

**Figure 10.** Diazepam adsorption isotherm for mineral clay; $T = 37.0 \pm 0.1$ °C; pH = $6.80 \pm 0.05$; S:L ratio = 1 g/L; ■ experimental data; and—Langmuir-fit adsorption curves.

In Table 8 the Langmuir equilibrium parameters for diazepam adsorption on activated charcoal and natural Na-montmorillonite are presented.

**Table 8.** Parameters $a_m$ and b for diazepam adsorption on activated charcoal and natural Na-montmorillonite.

| Sorbent | Langmuir Linear Equation | $a_m$ [mg g⁻¹] | $b$ [L·mg⁻¹] | $R^2$ |
|---|---|---|---|---|
| Activated charcoal | C/a = 0.0016C + 0.0285 | 611 | 0.058 | 0.993 |
| Natural Na-montmorillonite | C/a = 0.0769 C + 0.0651 | 13 | 1.179 | 0.999 |

### 3.4. Effect of pH on Diazepam Sorption

The surface charge of the sorbents and the pH of the bulk solution control the sorption of diazepam on the natural Na-montmorillonite and activated charcoal.

In Figures 11 and 12 is presented the influence of pH on diazepam sorption for natural Na-montmorillonite and active charcoal.

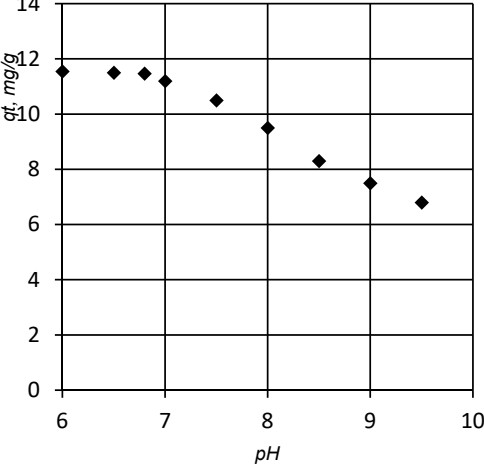

**Figure 11.** Influence of pH on diazepam sorption by natural Na-montmorillonite; $T = 37.0 \pm 0.1$ °C; $Ci = 18$mg/L; and S:L ratio = 1 g/L.

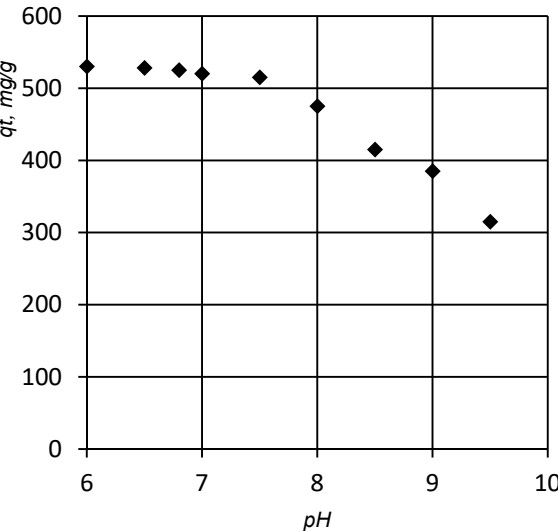

**Figure 12.** Influence of pH on diazepam sorption by activated charcoal; $T = 37.0 \pm 0.1$ °C; $Ci = 0.7$ mg/mL; and S:L ratio = 1 g/L.

### 3.5. The Influence of Ethanol Concentration on Diazepam Sorption

The sorption of diazepam from simulated intestinal fluid that contains different amounts of ethylic alcohol (96%, Merck) was studied. Langmuir isotherms were used to describe the experimental adsorption data for different concentration of ethanol in SIF. Figures 13 and 14 indicate that the amounts of diazepam uptake by both activated charcoal and natural Na-montmorillonite decrease in presence of ethanol.

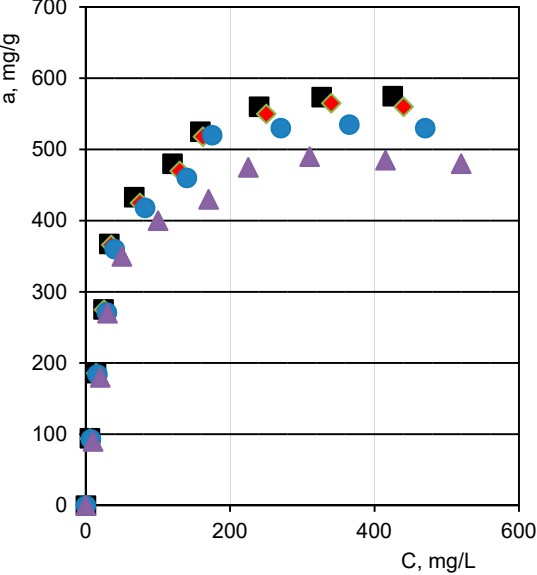

**Figure 13.** Diazepam adsorption isotherm on activated charcoal from simulated intestinal fluid (SIF) without and with ethanol pH = 6.80 ± 0.05; $T = 37.0 \pm 0.1$ °C; ■ SIF without ethanol; ◆ SIF with 5% ethanol; ● SIF with 10% ethanol; and ▲ SIF with 15% ethanol.

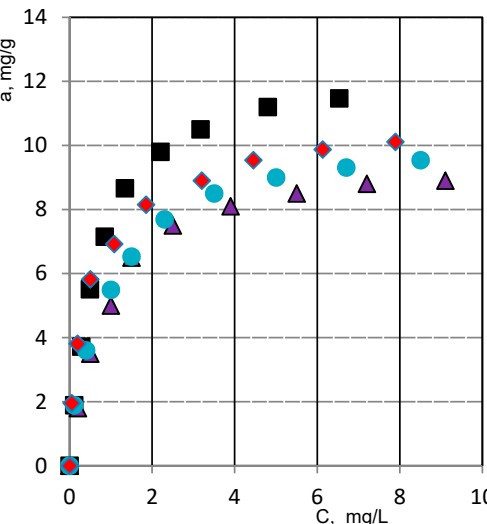

**Figure 14.** Diazepam adsorption isotherm on Na-montmorillonite from SIF without and with ethanol pH = 6.80 ± 0.05; $T$ = 37.0 ± 0.1 °C; ■ SIF without ethanol; ◆ SIF with 5% ethanol; ● SIF with 10% ethanol; ▲ SIF with 15% ethanol.

The maximum amounts, of diazepam adsorbed from SIF with and without ethanol are shown in Table 9.

**Table 9.** Maximum amount of diazepam adsorbed from SIF with and without ethanol.

|  | *a* (mg/g) | pH | Ethanol(% *v/v*) |
|---|---|---|---|
| Activated charcoal | 585 | 6.81 | 0 |
|  | 560 | 6.85 | 5 |
|  | 530 | 6.84 | 10 |
|  | 480 | 6.80 | 15 |
| Natural Na-montmorillonite | 11.5 | 6.85 | 0 |
|  | 10.1 | 6.82 | 5 |
|  | 9.3 | 6.80 | 10 |
|  | 8.7 | 6.82 | 15 |

## 4. Discussions

Specific surface area of the sorbents was determined, by applying the BET equation, as 1425.5 m²·g⁻¹ for activated charcoal and 112.5 m²·g⁻¹ for Na-montmorillonite.

The activated charcoal particle size varies in range 0–18 μm 93.1% of the particle being less 10 μm. Furthermore, activated charcoal pore size varies in range 10–100 Å over 68% being smaller than 25 Å.

It can be seen that 92% of natural Na-montmorillonite particles are smaller than 2 μm and the pore size of the mineral clay varies in range 10–300 Å about 35% being in range 10–25 Å.

The mineral clay contains important quantities of aluminum, silica, sodium, potassium, calcium and iron oxides and other elements in trace amounts in the form of oxides. The active charcoal is rich in surface functional groups C-O like in alcohols, phenols, acids, ethers and esters groups with well characteristics for adsorption. SEM analysis confirms that the pores on the activated charcoal and Na-montmorillonite surface were uniform in the micropore to mesopore size range, but the main difference of both adsorbents is their pore volume, which is nearly 10-times greater for the activated charcoal than for mineral clay.

For both sorbents, the diazepam sorption equilibrium was attained within 2 h, for all values of the drug SIF solutions initial concentrations. The amounts of diazepam uptake after 4 h were approximatively the same as the amounts sobbed after 2 h. Thus, we considered that drug adsorption equilibrium was attained in 2 h.

The amounts of diazepam uptake on solid phase increase while the sorbent particle size decreases.

Langmuir plots showed excellent coefficient of determination ($R^2$) for both sorbents. The maximum adsorption capacity for on activated charcoal and natural Na-montmorillonite are 611 mg/g and 13 mg/g, respectively.

The $pH_{ZPC}$ of natural Na-montmorillonite was determined to 7.3 while for activated charcoal to 7.6.

Below the $pH_{ZPC}$, the surface of the sorbents is positively charged and above the point is negative. Diazepam is a weakly basic or neutral drug [24]. The results exhibit that sorption of diazepam is higher below this point (Figures 11 and 12), due the neutrality of the drug which adsorbs by weak van der Waals forces, via an attraction of the positively charged surface sites at lower pH [7,23].

The decreases of sorption capacity above $pH_{ZPC}$ could be connected to the gradual ionization of diazepam in alkaline medium.

The presence of alcohol in SIF influences the diazepam sorption. The sorption of diazepam from simulated intestinal fluid that contains different amounts of ethylic alcohol (96%) is well described by Langmuir isotherms. It can be observed that increasing the ethanol concentration causes a decreasing of diazepam concentration in solid phase.

Some previous studies suggest that the water–ethanol solution is less polar than pure water [25] and thus, a solution containing ethanol compared to a pure aqueous solution makes drugs less adsorbable to activated charcoal [26].

Comparing Figures 13 and 14, one can see that the adsorption degree of diazepam on natural Na-montmorillonite is lower than on activated charcoal for the same concentration of ethanol. We appreciate, beside the fact that the water–ethanol solution is less polar than pure water, the ethanol–water solutions increased the swelling volume of this mineral clay. A higher concentration of ethanol increases mineral clay volume which determine decreasing of the pore volume and porosity [27].

## 5. Conclusions

This study shows that both activated charcoal and natural Na-montmorillonite can be used as antidote for diazepam overdose. As we expect, activated charcoal has a much higher sorption capacity than mineral clay that is explained by its greater pore volume. The Langmuir isotherm model provides a good description of the sorption of diazepam on the sorbents. Two hours represent enough time to reach diazepam sorption equilibrium. For both sorbents, adsorption of diazepam is higher below the $pH_{ZPC}$ of activated charcoal and natural Na-montmorillonite, due to the neutrality of the drug which is adsorbed by weak van der Waals forces of the positively charged surface sites.

The amounts of diazepam uptake by both activated charcoal and natural Na-montmorillonite decrease in the presence of ethanol. Increasing of the ethanol concentration causes a decreasing of the diazepam concentration in solid phase.

There is an increasing need for calculating the appropriate amount of activated carbon that must be administrated during acute poisoning, since overloading the patient with activated carbon is not free from side effects.

**Author Contributions:** Conceptualization, M.S. and D.S.S., methodology, I.S., software, I.-A.N.; validation, M.S. and D.S.S.; formal analysis, V.O.; investigation, M.S. and D.S.S.; resources, D.S.S. data curation, I.-A.N.; writing—original draft preparation, I.S.; writing—review and editing, I.-A.N.; supervision: M.S., V.O.; funding acquisition, D.S.S. All authors have read and agreed to the published version of the manuscript.

**Funding:** This research was funded by University Politehnica of Bucharest.

**Acknowledgments:** In this section you can acknowledge University Politehnica Bucharest that support the article publication.

**Conflicts of Interest:** The authors declare no conflict of interest. The funders had no role in the design of the study; in the collection, analyses, or interpretation of data; in the writing of the manuscript, or in the decision to publish the results.

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
