# Peer review of "Influence of Internal Structure of the Sorbents on Diazepam Sorption from Simulated Intestinal Fluid"

_applsci, doi:10.3390/app11031158_

Round 1

Reviewer 1 Report

The manuscript is of interest for the researchers who a dealing with drugs overdose in a practical sense. However, this manuscript should go through major changes to be accepted for publication.

My comments are mainly given in attached file. Some others are below.

Introduction

I think that it is rather strange, that the statistical data are taken from USA, not from the country of paper origin. The first part of introduction (lines 34-58) have to be changed, the reference list has to be renewed and improved (see att.1, I picked up just some references, without to go into details).

Line 62: “The surface area of the resulted material is about 1,000 – 2,000 m2 per gram”. I found some other data, for example: 766-815 m2 per gram (K.-W. Stahmer, M. Gerhold / Journal of Loss Prevention in the Process Industries 46 (2017) 177-184).

Materials and Methods (the rest of the comments are in the text, see attached file).

Line 98: quality of water has to be given.

Table 2.

1 line: Brunauer–Emmett–Teller (BET) is not a method, but the theory (or, at least, equation).

Line 118          2.3.1. Influence of initial drug concentrations

  1. It is known that the properties of sorbent are very much dependent on different conditions of carried out experiments, such as humidity, temperature, etc. It’ll be interesting to know about the preparation procedure of both activated carbon and Na-montmotillonite before the immersion to SIF solutions.
  2. Did you check the Nernst function of your glass electrode?
  3. It is known that the pH is temperature dependent, why it was measured at 20oC, not 37oC.
  4. The most important part of HPLC method is a type of chromatographic column. This information is missing. Please give the composition if eluent, flow rate, wave length, etc. How the column was calibrated? How the chromatograms are look like?

Line 165          2.6. The influence of ethanol on diazepam sorption

The use of binary solvent water-ethanol will change the experimental procedure: pH and HPLC measurements have to be described. The glass electrode has to be calibrated in each of the mixtures: 5%, 10% and 15% ethanol.

Results

Line 253          3.2.2. Influence of sorbent particle sizes

Two figures (fig.7 and Fig.8) are presented, nothing is written to describe them.

References for you help:

1.OPIOID-RELATED OVERDOSE IN PATIENTS WITH CONCURRENT USE OF OPIOIDS AND BENZODIAZEPINES: A NESTED CASE-CONTROL STUDY OF COMMERCIALLY INSURED PATIENTS

By: Alobaidi, A.Lee, T. A.

VALUE IN HEALTH  Volume: ‏ 23   Supplement: ‏ 1   Pages: ‏ S133-S133   Meeting Abstract: ‏ PDG24   Published: ‏ MAY 2020

2.Risk of Overdose with Exposure to Prescription Opioids, Benzodiazepines, and Non-benzodiazepine Sedative-Hypnotics in Adults: a Retrospective Cohort Study

By: Cho, JoanneSpence, Michele M.Niu, Fang; et al.

JOURNAL OF GENERAL INTERNAL MEDICINE  Volume: ‏ 35   Issue: ‏ 3   Pages: ‏ 696-703   Published: ‏ MAR 2020

3.Changes in Opioid-Involved Overdose Deaths by Opioid Type and Presence of Benzodiazepines, Cocaine, and Methamphetamine-25 States, July-December 2017 to January-June 2018

By: Gladden, R. MattO'Donnell, JulieMattson, Christine L.; et al.

MMWR-MORBIDITY AND MORTALITY WEEKLY REPORT  Volume: ‏ 68   Issue: ‏ 34   Pages: ‏ 737-744   Published: ‏ AUG 30 2019

4.Overdosing of benzodiazepines/z-drugs and falls in older adults: costs for the health system

By: Diaz-Gutierrez, M. J.Martinez-Cengotitabengoa, M.Bermudez-Ampudia, C.; et al.

EUROPEAN PSYCHIATRY  Volume: ‏ 56   Supplement: ‏ S   Pages: ‏ S174-S174   Meeting Abstract: ‏ E-PP0737   Published: ‏ APR 2019

5.Overdosing of benzodiazepines/Z-drugs and falls in older adults: Costs for the health system

By: Jose Diaz-Gutierrez, MariaMartinez-Cengotitabengoa, MonicaBermudez-Ampudia, Cristina; et al.

EXPERIMENTAL GERONTOLOGY  Volume: ‏ 110   Pages: ‏ 42-45   Published: ‏ SEP 2018

6.POLYSUBSTANCE ABUSE: THE OVERLOOKED FACTOR OF BENZODIAZEPINES IN OPIOID OVERDOSE IN HARRIS COUNTY

By: Lacey, R. C.Dean, E.Onyeigo, S.; et al.

RESEARCH IN SOCIAL & ADMINISTRATIVE PHARMACY  Volume: ‏ 14   Issue: ‏ 6   Pages: ‏ E20-E20   Published: ‏ JUN 2018

7.Problems with the prescription of benzodiazepines in Japan. Evaluation of suicide attempts by overdosing

By: Akahane, A.Matsumura, K.Ikebuchi, E.

Conference: 29th CINP World Congress of Neuropsychopharmacology Location: ‏ Vancouver, CANADA Date: ‏ JUN 22-26, 2014
Sponsor(s): ‏CINP

INTERNATIONAL JOURNAL OF NEUROPSYCHOPHARMACOLOGY  Volume: ‏ 17   Special Issue: ‏ 1   Pages: ‏ 45-45   Meeting Abstract: ‏ P-01-001   Published: ‏ JUN 2014

 8.Benzodiazepines: a major component in unintentional prescription drug overdoses with opioid analgesics.

By: Jann, MichaelKennedy, William KlughLopez, Gaylord

Journal of pharmacy practice  Volume: ‏ 27   Issue: ‏ 1   Pages: ‏ 5-16   Published: ‏ 2014-Feb

9.Presence of lethal or non-lethal levels of mood stabilizers, atypical antipsychotics, antidepressants, and benzodiazepines at time of death in bipolar disorder overdose suicides

By: Schaffer, A.Sinyor, M.Goldstein, B. I.; et al.

Conference: 10th International Conference on Bipolar Disorder of the International-Society-for-Bipolar-Disorders Location: ‏ Miami Beach, FL Date: ‏ JUN 13-16, 2013
Sponsor(s): ‏Int Soc Bipolar Disorders

BIPOLAR DISORDERS  Volume: ‏ 15   Special Issue: ‏ SI   Supplement: ‏ 1   Pages: ‏ 90-90   Published: ‏ JUN 2013

10.Non-fatal overdose of duloxetine in combination with other antidepressants and benzodiazepines

By: Menchetti, MarcoGozzi, Beatrice FerrariSaracino, Maria Addolorata; et al.

WORLD JOURNAL OF BIOLOGICAL PSYCHIATRY  Volume: ‏ 10   Issue: ‏ 4   Pages: ‏ 385-389   Part: ‏ 2   Published: ‏ 2009

11.Clinically relevant anterograde amnesia and its relationship with blood levels of benzodiazepines in suicide attempters who took an overdose

By: Verwey, B; Muntendam, A; Ensing, K; et al.

PROGRESS IN NEURO-PSYCHOPHARMACOLOGY & BIOLOGICAL PSYCHIATRY  Volume: ‏ 29   Issue: ‏ 1   Pages: ‏ 47-53   Published: ‏ JAN 2005

12.Alprazolam is relatively more toxic than other benzodiazepines in overdose

By: Isbister, GK; O'Regan, L; Sibbritt, D; et al.

BRITISH JOURNAL OF CLINICAL PHARMACOLOGY  Volume: ‏ 58   Issue: ‏ 1   Pages: ‏ 88-95   Published: ‏ JUL 2004

13.Alprazolam is relatively more toxic than other benzodiazepines in overdose.

By: Isbister, G. K.; O'Regan, L.; Sibbritt, D.; et al.

Conference: Annual Meeting of the North American Congress of Clinical Toxicology Location: ‏ Chicago, IL, USA Date: ‏ September 04-09, 2003
Sponsor(s): ‏North American Congress of Clinical Toxicology

Journal of Toxicology Clinical Toxicology  Volume: ‏ 41   Issue: ‏ 5   Pages: ‏ 715   Published: ‏ August 2003

14.RELATIVE TOXICITY OF BENZODIAZEPINES IN OVERDOSE

By: BUCKLEY, NADAWSON, AHWHYTE, IM; et al.

BMJ-BRITISH MEDICAL JOURNAL  Volume: ‏ 310   Issue: ‏ 6974   Pages: ‏ 219-221   Published: ‏ JAN 28 1995

15.IMPACT OF TOXICOLOGY SCREENS IN THE DIAGNOSIS OF A SUSPECTED OVERDOSE - SALICYLATES, TRICYCLIC ANTIDEPRESSANTS, AND BENZODIAZEPINES

By: JAMMEHDIABADI, MTIERNEY, M

VETERINARY AND HUMAN TOXICOLOGY  Volume: ‏ 33   Issue: ‏ 1   Pages: ‏ 40-43   Published: ‏ FEB 1991

Author Response

Dear reviewer, thank you for your observations.

We have made the corrections. 

Reviewer 2 Report

Dear Authors,

Please find attached my comments on your article. It may be interesting but some revision should be made and some parts of manuscript have to be explained or improved.

Author Response

(The authors gave the same response as above.)

Reviewer 3 Report

The topic and work of your article is interestingly presented. But there are two main problems:

1) In Table 7 the Parameters of the Langmuir fit are determined wrongly. The values for am correspond nearly to Figures 9 and 10 but the Langmuir equations in the second column do not fit. Furthermore, if  +/- is written a corresponding number has to be inserted.

2) The last sentence of the discuccsion has to be revised:

In [25] is written that with increasing EtOH-concentration the clay swelling increases. Due to that, the permability of the clay is reduced. Now, it has to be distinguished between the different scales: for permeability processes greater cavities are essential. The determined pore volumes in this article correspond to the region of micro- and smaller mesopores, in which the adsorption takes place by diffusional processes but permeability does no play any role.

It has to be emphasized that the main difference of both adsorbents is their pore volume, which is nearly 10-times greater for the activated charcoal than for the montmorillonite. Therefore it is obvious that the Adsorption capacity of the activated charcoal is greater than of the montmorillonite.

Now, minor mistakes/changes are listed:

line 57: ..depression. It must be noted that …..

line 64: ….micropores to macropores. The Adsorption process...

line 70/71: ...that have made it possible....

line 75: remove one dot.

line 78: … responsible….

Table 4: FTIR: Fourier transform infrared.....:IR spectrum of mineral....

line 112:

line 115: missing the dot

line 124: missing dot

line 126: no comma in front of "and": e.g. 0.5 and 0.7 g/…..

line 148: ...concentrations ranged from …..

line 159: ...SIF solution concentration...

line 160: ….drug concentration was …..; one bracket is missing

line 166: ...Ethanol on Diazepam....

line 170: ….solid:liquid ratio was 1 g/L.

line 177: ...in the next four tables.

line 208: ….at 1162 cm_1.....

line 215:......technique is presented.

line 231: ...in the range of ….

line 232-234: The technique confirmed fluffy or open-textured Na-montmirillonite which is presumably ….

line 237/238: ...confirms the presence of micropores…… that were identified by N2 Adsorption technique.

Figure 5 & 6: A Frame around each figure would enhance the clarity in the presentation. In Fig. 6 the labe of the y-axis is not visible.

line 254: ...particle size was investigated, as shown in …..

Figure 7 & 8: signature below the figures: ….particle sizes

line 261/262: the word "metal" has to be changed !

Figure 10: The word "Figure" and the unit of the x-axis have to be changed

and between the pH-value and the temperature T is a wrong sign

Table 7: see the comment at the beginning

line 276: Sorption and in the signature of Figures 11 & 12, too.

line 279: A heading like 3.5: Effect of EtOH-concentration …..is missing

line 281:.... different concentration….

line 281: Number of Figures is wrong.

line 289/290: Keep the unit the specific surface together in one line.

line 291-293: The activated charcoal particle size varies in range of 0-18 μm, 93.1% of the particle beeing less than 10 μm. Also activated charcoal pore size varies in range of 10-100 A, over 65% being smaller than 25 A. It can be seen that 92 % of natural Na-montmorillonite particles are smaller….

line 298: ...SEM analysis confirms that the pores on ...

line 300: .. range but the pore volume is very different of both samples.

line 309: ...determined to 7.3 while for activated carbon  to….

line 316: ….influences the Diazepam...

line 317: ..is well described by the Langmuir equation. "Table 8" ??

line 319: ......….concentration in the adsorbed Phase? instead of solid phase?

line 320 : remove the dot behind the bracket.

line 326/327: This sentence has to be revised corresponding to the comment at the beginning.

line 338/339: Revision of this sentence is necessary.

line 375:...."gut" decontamination?

line 387: ...Activated Carbon,  not italic

line 394: : space between "carbons from…"

Author Response

Dear reviewer, thank you for your observations.

We have made the following correction:

Round 2

Reviewer 1 Report

The manuscript have to be improved to be published.

Author Response

Dear reviewer, thank you for your observations.

We have made the  correction attached. 

Reviewer 2 Report

Dear Authors,

I am glad to see your improved manuscript. Now I recommend to accept it.

Author Response

Dear reviewer, thank you for taking this  decision.

Reviewer 3 Report

After the correction of only minor spelling mistakes the article is ready for publication.

line 169: ...drug concentration was....

line 237: ..vibrations .... support the ….

line 277: In the unit is a wrong blank space between the "m" and the "g".

line  278: …. linearisation….

line 297: … different concentration ...

line 309: ….. Na-montmorillonite particles are smaller….

line 333: …. diazepam ...

line 334: …..well described by ….

Author Response

Dear reviewer, thank you for your observations.

We have made all corrections that you suggested.
